# A *TM*_01_-*TE*_11_ Circular Waveguide Mode Converter on the Basis of Dielectric Filling

**DOI:** 10.3390/mi16050585

**Published:** 2025-05-16

**Authors:** Zibin Weng, Ziming Lv, Liupeng Zan, Sihan Xiao, Chen Liang

**Affiliations:** The National Key Laboratory of Radar Detection and Sensing, Xidian University, Xi’an 710071, China; 23021211267@stu.xidian.edu.cn (Z.L.); 23021211951@stu.xidian.edu.cn (L.Z.); 23021211373@stu.xidian.edu.cn (S.X.); liangchen@stu.xidian.edu.cn (C.L.)

**Keywords:** media filling, *TM*_01_-*TE*_11_, mode conversion, circular waveguide, waveguide mode matching, high efficiency

## Abstract

In this paper, a dielectric-filled circular waveguide ${TM}_{01}$*-*${TE}_{11}$ mode converter is proposed, which has high conversion efficiency and a wide operating bandwidth. Filling the circular waveguide with dielectric material changes the local propagation characteristics, thus achieving a phase difference between the ${TE}_{11}$ modes in the two halves of the circular waveguide during propagation. This, in turn, facilitates the completion of mode conversion with high efficiency. Compared with the conventional radial dielectric plate, this paper improves the method of filling the dielectric inside the circular waveguide by transforming it into a coaxial structure. This is followed by the incorporation of a radial dielectric plate, a modification that has been proven to enhance the conversion efficiency and extend the operational bandwidth. The mode converter operates at 9.7 GHz, and when the dielectric filler material is polytetrafluoroethylene (PTFE), both simulation and practical studies are carried out. The simulation results demonstrate that the maximum conversion efficiency of this mode converter is 99.2%, and the bandwidth with conversion efficiency greater than 90% is nearly 21.1%. The maximum conversion efficiency in the actual test is essentially consistent with the simulation results. The validity of the design scheme of this converter and the accuracy of the simulation study are demonstrated.

## 1. Introduction

Mode conversion is one of the core technologies in microwave and millimeter wave systems. To achieve multi-mode beam switching and accurate detection of different targets, waveguide mode conversion is widely used in radar systems [1,2,3,4,5,6]. Mode converters are also widely used in the transmission of communications and play an important role in the processing of signals and in the conversion of multi-mode signals [7,8]. In circular waveguides, the effective conversion between ${TM}_{01}$ and ${TE}_{11}$ modes constitutes a pivotal application of mode converters, which are extensively employed in microwave and millimeter wave systems. While existing methodologies for waveguide mode conversion exhibit distinct operational advantages, it is imperative to acknowledge the inherent limitations fundamentally associated with each approach when evaluating their practical implementation in microwave systems [9,10,11,12,13]. For example, the serpentine waveguide converter based on multi-periodic perturbation has a high power capacity. However, its length is too long, making it difficult to install in practical applications [1,12]. Alternatively, a metal-inserted mode converter can be used, but this greatly increases the complexity of manufacturing waveguide devices and significantly reduces their operating bandwidth [4,14,15]. Meanwhile, the input and output ports of the mode converter with right-angle structures do not conform to the coaxial configuration; therefore, its application in compact systems is limited [16].

To overcome these shortcomings, this paper proposes the modification oof the propagation characteristics of the local region of the circular waveguide by introducing a dielectric material inside the waveguide to achieve an efficient conversion between the ${TM}_{01}$ mode and the ${TE}_{11}$ mode. In comparison with the conventional medium filling method [3,17,18,19], this design effectively suppresses the generation and excitation of spurious modes by further optimizing the parameter selection of the filling medium. This design has been shown to reduce the complexity of the structure, thereby facilitating a significant simplification of the overall structure, resulting in a more compact and shorter design. Furthermore, it has been demonstrated that this design is superior in performance. It is evident that the design is capable of attaining higher conversion efficiency and a wider conversion bandwidth while exhibiting a balanced approach between high performance and compactness. This is indicative of its superior overall performance.

## 2. Mode Converter Design Methodology and Parameter Selection

The design concept of the dielectric-filled circular waveguide mode converter is outlined in this paper.

Due to some of the limitations of conventional mode transformers, and because media filling eliminates the need for complex mechanical structures and simplifies the process, media filling has been chosen as the most efficient way to transform from ${TM}_{01}$ mode to $\;{TE}_{11}$ mode. The ${TM}_{01}$ mode has an axisymmetric electric field distribution. The ${TE}_{11}$ mode requires the electric field to form a transverse distribution across the waveguide cross section. To achieve this conversion from ${TM}_{01}$ mode to ${TE}_{11}$ mode, this design has been refined and improved from the semicircular media filling method to the radial media plate filling method, as well as to the coaxial structure conversion and radial media plate loading method, achieving efficient mode switching. Eventually, by filling the circular waveguide with a medium, converting it into a coaxial structure, and dividing it into two semicircular waveguides using a metal plate, the flexible manipulation of the modes of the waveguide is achieved.

In the instance of a ${TM}_{01}$ mode being transmitted in a circular waveguide, it undergoes conversion to a ${TE}_{11}$ mode in a semicircular waveguide. Subsequently, a radial dielectric plate is inserted into one of the semicircular waveguides, and the dielectric constants and fill lengths of the dielectric materials are precisely tuned. As a result, the ${TM}_{01}$ modes in the two semicircular waveguides are made unequal in their propagation coefficients, and the ${TE}_{11}$ modes are realized to produce a half-periodic difference in the phase difference in the two semicircular waveguides [20,21,22,23]. The final stage of the conversion process involves the transformation of the ${TE}_{11}$ mode into the target mode, which is facilitated by a transition waveguide segment. This design facilitates efficient mode conversion.

Figure 1 provides a visual representation of the fundamental configuration of the dielectric-filled circular waveguide mode converter. Table 1 shows the variables associated with the mode converters filled by a dielectric of this design.

When the circular waveguide ${TM}_{01}$ mode is transmitted at the A end, the BC segment utilizes a filled dielectric to generate a half-cycle phase discrepancy between the ${TE}_{11}$ modes within a double half-circular waveguide. The CD section, in this configuration, assumes the role of a transition waveguide, facilitating the conversion of coaxial line ${TE}_{11}$ modes into their counterpart ${TE}_{11}$ modes within the circular waveguide.

The coefficient of propagation of an electromagnetic wave in a semicircular waveguide is obtained using Equation (1).(1)$$\beta =\sqrt{{\left(2\pi f\right)}^{2}\mu \varepsilon_{0}\varepsilon_{r}-{k_{c}}^{2}}$$

In this text, ‘$k_{c}$’ denotes the cut-off wave number. It is intimately linked to several factors, namely the inner and outer radius of the circular waveguide, the angle of the center of the circle, and the mode of propagation within the waveguide.

It is evident that the permittivity ($\varepsilon_{r}$) of the two semicircular waveguides is not uniform. This results in the propagation coefficients (*β*_1_, *β*_2_) of the two waveguides propagating the same modes show that they are different when the length of the filled dielectric material (*L*) satisfies [17,18](2)$$(\beta_{1}-\beta_{2})L=\pi$$

The medium of the BC section provides the necessary half-cycle phase difference for ${TE}_{11}$ mode transitions in the double semicircular waveguide. At point C, the direction of the electric field in the two semicircular waveguides forms a phase difference of exactly 180°. The transition from the semicircular waveguide to the circular waveguide then occurs after the CD section, thus completing the conversion of the circular waveguide from ${TM}_{01}$ to ${TE}_{11}$. The conversion process is illustrated in Figure 2.

The length and dielectric constant of the filling medium have a direct effect on the performance of the converter. Consequently, this design is focused on the rational selection of the medium material and determining its appropriate length. The determination of these parameters can be achieved through the utilization of Equations (1) and (2).

The length $L_{BC}$ of the filling medium [17,24] is shown in Equation (3).(3)$$L_{BC}=\pi /\left(\sqrt{{\left(2\pi f_{0}\right)}^{2}\mu \varepsilon_{0}\varepsilon_{r}-{k_{c}}^{2}}-\sqrt{{\left(2\pi f_{0}\right)}^{2}\mu \varepsilon_{0}-{k_{c}}^{2}}\right)$$

According to the phase superposition condition of the electromagnetic field, in theory, the ${TE}_{11}$ modes in the top and bottom half of the circular waveguide with a phase difference of π should have a conversion efficiency of 90% in the synthesis of the ${TE}_{11}$ modes in the circular waveguide, which is required to satisfy Equation (4).(4)$$\cos^{2}\theta \cos^{2}(\pi \pm \theta )=0.9$$

The selection of a suitable dielectric material is of paramount importance in ensuring optimum converter performance, both in terms of high conversion efficiency and the wide operating bandwidth. According to the bandwidth constraints within the operating band, in order to achieve an efficiency level exceeding 90% of the bandwidth, the converter conversion efficiency must satisfy the following Equation (5):(5)$$\pi -\theta <(\beta_{1}(\varepsilon_{r},f)-\beta_{2}(f))\times L_{BC}(\varepsilon_{r})<\pi +\theta$$

The above Equation (5) analysis demonstrates that the operating bandwidth of the converter is directly proportional to the magnitude of the dielectric constant of the dielectric material. However, this also means that a longer dielectric material is required, which is not favorable for compact system applications. Consequently, a trade-off exists between the length and the operating bandwidth of the converter, and it is not possible to achieve a reduction in size or an increase in the operating bandwidth simply by selecting the dielectric material.

In order to achieve a relatively wide operating bandwidth, it was decided that the optimum filler medium material would be PTFE with a dielectric constant of 2.1. The loss tangent is 0.0002. Because of its low dielectric loss, good thermal stability, and low relative dielectric constant, PTFE is well suited for microwave and RF applications.

## 3. Simulation Verification and Real Measurement

The simulation verification and optimization of the mode converter were carried out using HFSS EM simulation software. The input and output ports are excited by wave ports, with the input ports being configured to the desired ${TM}_{01}$ mode excitation. The thickness of the metal plate in the center of the circular waveguide is 2 mm. The length of the metal plate needs to match the length of the dielectric plate *L* to ensure mode conversion efficiency.

The thickness of the dielectric plate affects the propagation constant. If the thickness differs from the theoretically calculated value, the propagation constant difference will not match the theoretical value, resulting in phase shift errors. At the same time, a dielectric plate that is too thick may cause large electromagnetic wave reflections, reducing the conversion efficiency. Therefore, the radial dielectric thickness parameter is scanned in the simulation to find the optimum thickness for a given waveguide and operating frequency. The thickness is finally decided to be 2.07 mm.

The impact of employing varied distributions for populating the media within the mode converter on the conversion efficiency is demonstrated in Figure 3. Figure 3 shows a comparison of the conversion efficiencies obtained for these filling methods.

As can be seen from the comparison in Figure 3, the use of the (c) method of filling the dielectric has a higher mode conversion efficiency compared to the (a) method of adding the dielectric plate directly, and also its bandwidth is better than in the (a) and (b) methods.

Through the design method in (a), the direct addition of dielectric plates, although simple in structure, has obvious limitations in efficiency, bandwidth, and other indicators, and phase matching needs to rely on the optimization of complex dielectric parameters. Methods (b) and (c) have been shown to achieve coaxial-like functionality through the use of structural segmentation with dielectric filling. In this configuration, the semicircular waveguides filled with a medium and air correspond to the dielectric layer and gap of the coaxial structure, respectively. The electromagnetic field distribution is modulated by the difference in the dielectric constants. Meanwhile, method (c) incorporates a transition waveguide, characterized by a CD segment, in contrast to method (b). The waveguide transition segment is engineered to facilitate a seamless transformation of the field distribution through the utilization of geometrical asymptotics, thereby effectively suppressing higher-order mode excitations. The approximate coaxial structure has been demonstrated to enhance the phase matching and to control the propagation constants in the double semicircular waveguide with a high degree of precision, thus satisfying the condition of 180° phase difference as a way to achieve efficient mode conversion. The design can effectively expand the bandwidth by optimizing the dielectric material parameters, which causes the mode converter to have high efficiency and wide bandwidth. Therefore, it was finally decided to use method (c) to fill the medium.

Figure 4 shows the process by which the efficiency of each mode varies with frequency during the conversion of the mode converter.

As can be seen from Figure 4, the highest conversion efficiency of the mode converter is 99.2% near 9.7 GHz, and the bandwidth where the conversion efficiency is 90% or more is 2.05 GHz, which is approximately 21.1%. The high performance of this mode converter is demonstrated by the simulation results.

This design is also compared with the existing mode converter in terms of bandwidth and conversion efficiency, and the results are shown in Table 2.

As shown in Table 2, the design achieves a relative bandwidth of 21.1% at a 9.7 GHz center frequency with a conversion rate of >90% and a maximum conversion efficiency of 99.2%. This is better than most referenced studies, such as the 19.2% relative bandwidth in reference [26] and the highest conversion efficiency in references [4,17,27]. The optimization of the fill medium reduces the frequency sensitivity, resulting in an increase in the relative bandwidth.

The electric field inside the converter during the mode conversion process is shown in Figure 5.

Figure 5a shows that the ${TE}_{11}$ modes of the upper and lower semicircular waveguides produce a half-cycle phase difference in the filled dielectric section. Figure 5c shows that the output mode of the mode converter is indeed the ${TE}_{11}$ mode.

The actual measurements were performed on a Vector Network Analyzer to determine the conversion efficiency and broadband characteristics of the mode converter by measuring $S_{21}$ at the output port. Figure 6 shows a picture of the practical test. Figure 6a shows the complete test system. Figure 6b shows the exciter plus mode converter. It consists of a ${TM}_{01}$ exciter, a ${TE}_{11}$ exciter, and the mode converter designed in this paper.

Figure 7 shows the actual tested $S_{21}$ of the mode converter compared with the simulation results.

Although the $S_{21}$ parameter curves tested in the experiment are slightly different from the simulation results, they are basically the same at the center frequency point. This indicates the correctness of the design idea of this converter and the simulation results. This mode converter can achieve higher conversion efficiency and wider conversion bandwidth and has the advantages of small size, easy processing, coaxial input and output, and so on.

## 4. Conclusions

In this paper, a dielectric-filled circular waveguide ${TM}_{01}$ to ${TE}_{11}$ mode converter is proposed, which has high conversion efficiency and wide operating bandwidth. The paper analyzes the effects of different ways of filling the medium on the mode conversion efficiency, finding that the filling medium first converts the circular waveguide into a coax structure and the addition of the radial medium plate significantly improves the conversion efficiency in the simulation and effectively extends the working bandwidth. The highest conversion efficiency of the mode converter is about 99.2% near 9.7 GHz, and the bandwidth at which the conversion efficiency exceeds 90% is about 21.1%, as verified by practical experiments, and the simulated and measured results are basically in agreement. In addition, the mode converter has the advantages of small size, ease of implementation, and coaxiality of the input and output ports.

The structure of this design does not require bending and can be directly integrated into a linear microwave system. This reduces the complexity of the system. In addition, the material, length, and thickness of the dielectric plate can be adjusted, providing a flexible means of optimization and design space for different frequency bands.

This design achieves a significant improvement in the bandwidth, efficiency, and ease of fabrication of the ${TM}_{01}$ to ${TE}_{11}$ mode converter through an innovative dielectric loading structure combined with a phase-matching method, guiding future compact and microwave systems.

## Figures and Tables

**Figure 1 micromachines-16-00585-f001:**
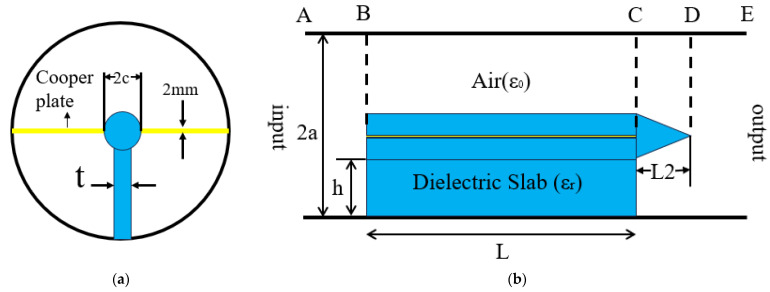
Structure of media-filled mode converter: (**a**) front view; (**b**) side view.

**Figure 2 micromachines-16-00585-f002:**
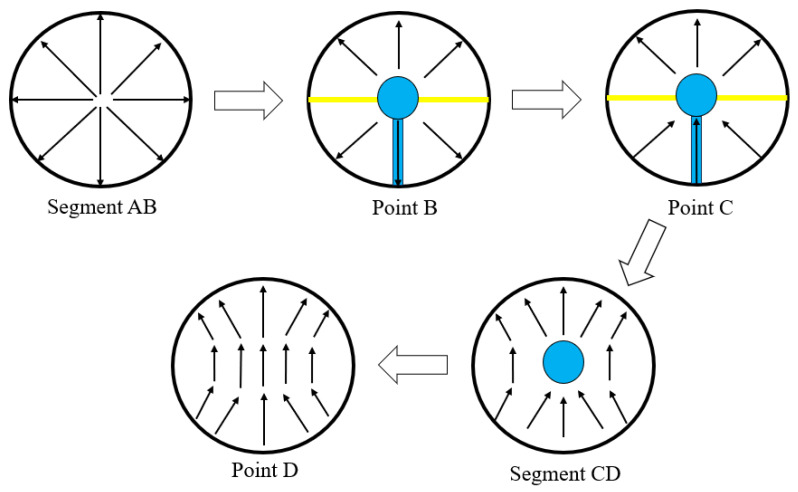
The basic process of circular waveguide mode conversion.

**Figure 3 micromachines-16-00585-f003:**
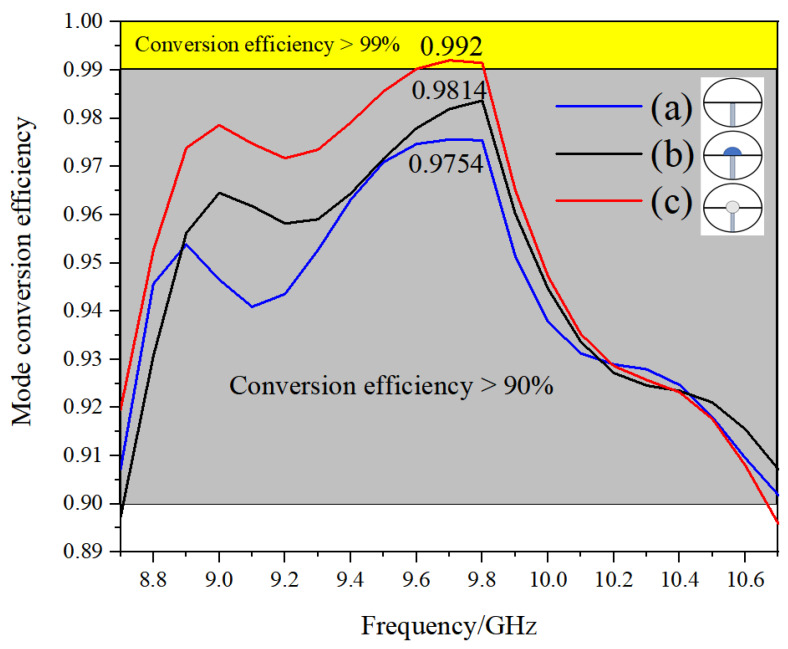
Effect of using different ways to fill the medium on conversion efficiency.

**Figure 4 micromachines-16-00585-f004:**
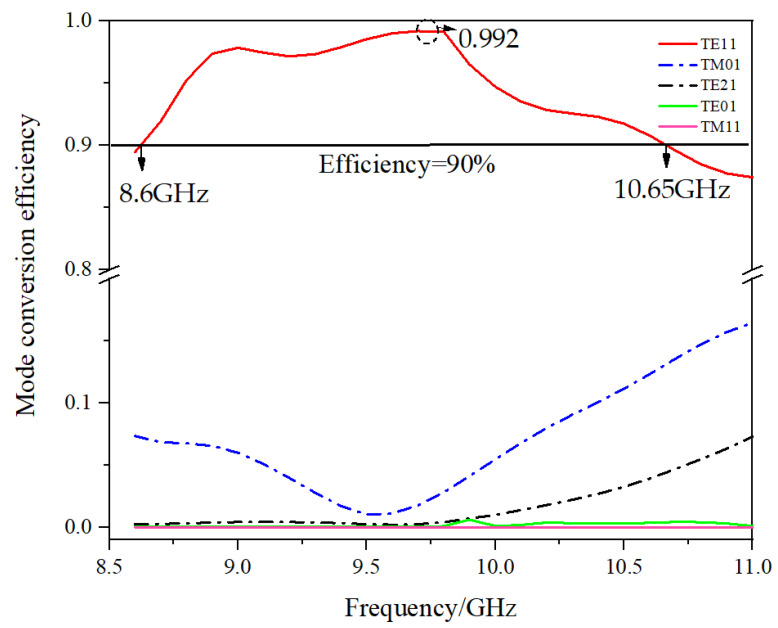
Variation in the efficiency of each mode with frequency during the conversion process.

**Figure 5 micromachines-16-00585-f005:**
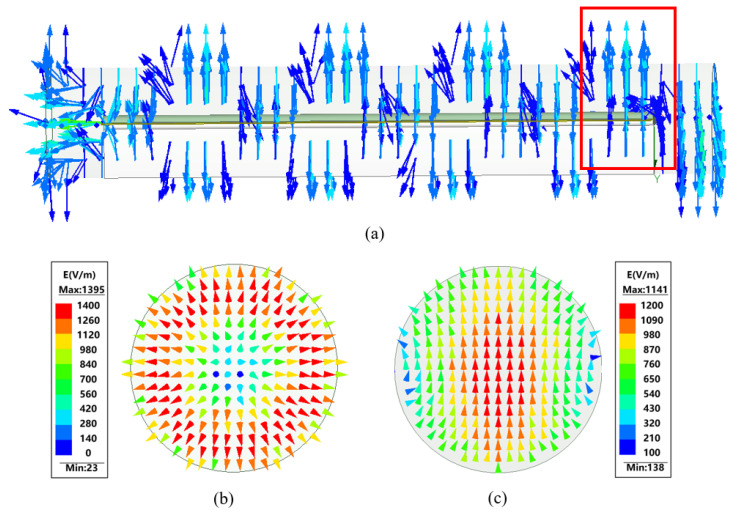
Converter internal electric field transformation. (**a**) Internal electric field distribution. (The red box indicates that the mode conversion is complete at this point.) (**b**) Input port mode. (**c**) Output port mode.

**Figure 6 micromachines-16-00585-f006:**
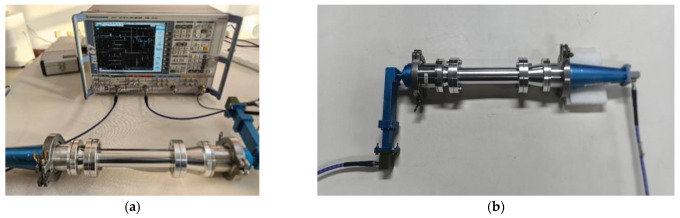
Practical test. (**a**) Overall test chart. (**b**) Mode converter object.

**Figure 7 micromachines-16-00585-f007:**
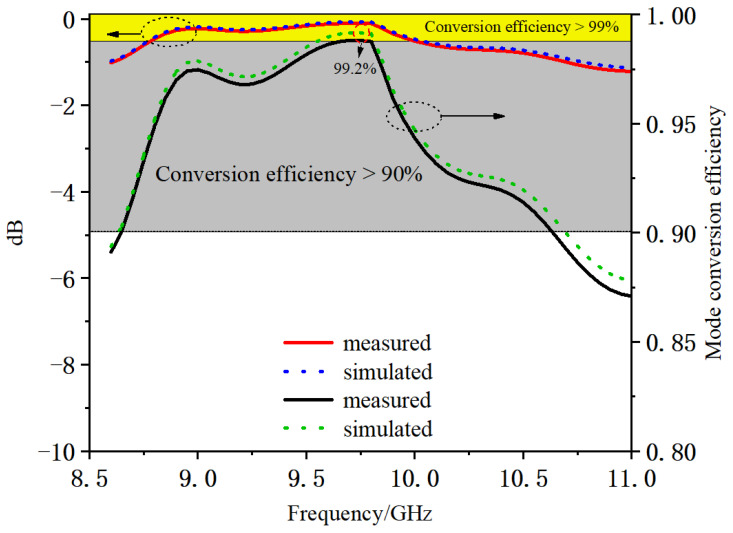
Comparison of measured and simulated results. (The direction of the arrow indicates the Y-axis of each data reference.)

**Table 1 micromachines-16-00585-t001:** The variables of the mode converter.

Symbol	L/mm	a/mm	t/mm	c/mm	h/mm	L2/mm
**Explanation**	Length of media plate	Radius	Thickness of the media plate	Radius of coaxial media structures	Height of media plate	Length of transition section
**Value**	149	15	2.07	2.5	14	5.65

**Table 2 micromachines-16-00585-t002:** Comparison results of this design with existing mode converters.

Design	Operating Frequency (GHz)	Radius (λ)	Bandwidth (%)	Max. Efficiency (%)
[4]	4	0.85	12.2	99
[14]	3.1	0.48	9.1	\
[17]	3	0.9	16.8	98.9
[25]	1.7	\	18.3	95
[26]	10.4	0.46	19.2	\
[27]	8.4	0.91	13	98.5
This work (c)	9.7	0.5	21.1	99.2

## Data Availability

All data generated or analyzed during this study are included in this manuscript. There are no additional data or datasets beyond what is presented in the manuscript.

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
