# Peer review of "A *TM*_01_-*TE*_11_ Circular Waveguide Mode Converter on the Basis of Dielectric Filling"

_micromachines, 2025, doi:10.3390/mi16050585_

Round 1
Reviewer 1 Report
Comments and Suggestions for Authors
In this work, the authors propose a dielectric-filled circular waveguide TM01-TE11 mode converter with a broad operational bandwidth and high conversion efficiency.
Comments
- How does the variation in dielectric plate thickness influence the propagation constant and mode conversion efficiency in the circular waveguide, and what simulation strategy is used in HFSS to determine the optimal thickness?
- What is the impact of different dielectric media distributions inside the mode converter on the mode conversion efficiency, as observed in the HFSS simulation results shown in Figure 3?
Reviewer 2 Report
Comments and Suggestions for Authors
A dielectric-filled circular waveguide TM01 - TE11 mode converter is proposed. The characterization is in agreement with simulation. There are some point to be improved:
- The design approach should be better illustrated till the identification of the internal final geometry. The logical flow showing how it has been refined should be reported step by step
- If possible a circuital model of the main parts of the converter could be provided
- The novelty should be better evidenced
Reviewer 3 Report
Comments and Suggestions for Authors 1, Could the author provide a table for the variables in figure 1? for example ,what is t, what is value of c, a, L, h, L2 and so on? So that readers could be easier to understand your paper. And your paper could also be more informative. For example, I am a little confuse about the line 143, what does the value of 2.07 mm stand for? 2, In line 94-96, are there some mistakes in the expression "and the electric field direction in the two semicircular waveguides is exactly opposite when at point C" . Obviously, at point B, electric filed are opposite, while at point C, they are the same. Please modify the expression mistake here. 3, In lines 174-176, could the author please also describe figure (b) to make the structure of the paragraph better? 4, Please improve the style of your English. Please ask a native English speaker to modify the paper. 5, The paper is short, maybe the article type "communication" could be better. " Comments on the Quality of English LanguagePlease improve the style of your English. Please ask a native English speaker to modify the paper.
